# Learning Spatial Common Sense with Geometry-Aware Recurrent Networks

**Hsiao-Yu Fish Tung**[1]* **Ricson Cheng** [2]*† **Katerina Fragkiadaki** [1]
[1]Carnegie Mellon University    [2]Uber Advanced Technologies Group
{htung,katef}@cs.cmu.edu, ricsonc@uber.com

## Abstract

We integrate two powerful ideas, geometry and deep visual representation learning, into recurrent network architectures for mobile visual scene understanding. The proposed networks learn to "lift" 2D visual features and integrate them over time into latent 3D feature maps of the scene. They are equipped with differentiable geometric operations, such as projection, unprojection, egomotion stabilization, in order to compute a geometrically-consistent mapping between the world scene and their 3D latent feature space. We train the proposed architectures to predict novel image views given short frame sequences as input. Their predictions strongly generalize to scenes with a novel number of objects, appearances and configurations, and greatly outperform predictions of previous works that do not consider egomotion stabilization or a space-aware latent feature space. Our experiments suggest the proposed space-aware latent feature arrangement and egomotion-stabilized convolutions are essential architectural choices for spatial common sense to emerge in artificial embodied visual agents.

## 1 Introduction

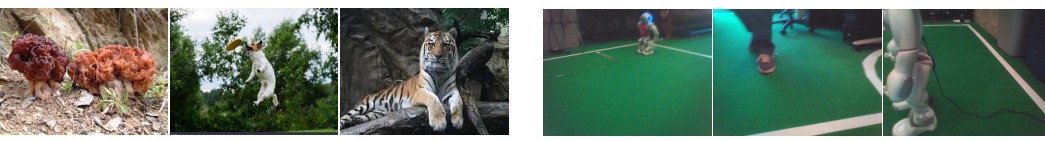

Figure 1: **Internet vision versus robotic vision**. Pictures taken by humans (left) (and uploaded on the web) are the *output* of visual perception of a well-trained agent, the human photographer. The content is skillfully framed and the objects appear in canonical scales and poses. Pictures taken by mobile agents, such as a NAO robot during a robot soccer game (right), are the *input* to such visual perception. The objects are often partially occluded and appear in a wide variety of locations, scales and poses. We present recurrent neural architectures for the latter, that integrate visual information over time to piece together the visual story of the scene.

Current state-of-the-art visual systems (5) accurately detect object categories that are rare and unfamiliar for many of us, such as *gyromitra*, a particular genus of mushroom (Figure 1 left). Yet, they neglect the basic principles of object permanence or spatial awareness that a one-year-old child has developed: once the camera turns away, or a person walks in front of the gyromitra, its detection disappears and it is replaced by the objects detected in the new visual frame. We believe the ability of current visual systems to detect rare and exquisite object categories and their inability to carry out elementary spatial reasoning is due to the fact that they are trained to *label object categories* from *static Internet photos* (contained in ImageNet and COCO datasets) using a *single frame* as input. Our overexposure to Internet photos makes us forget how pictures captured by mobile agents look. Consider Figure 1. Internet photos are skillfully captured by human photographers, are well framed and show objects unoccluded, in canonical locations, scales and poses (left). Instead, photos captured by NAO robots during a soccer game show objects in a wide variety of scales, poses, locations,

---

*Indicates equal contribution
†Work done while at CMU

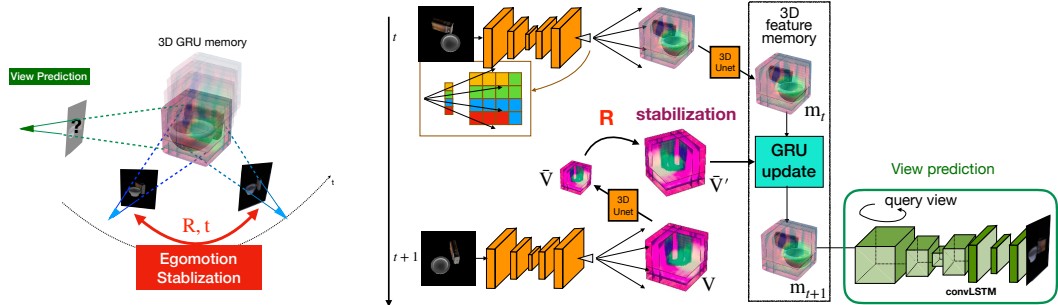

Figure 2: Left: **Geometry-aware Recurrent Neural Networks (GRNNs)** integrate visual information over time in a 3D geometrically-consistent GRU memory of the visual scene. Right: Neural components of GRNNs. RGB images are fed into a 2D U-net, the resulting deep features are unprojected to 4D tensors, fed into a 3D U-net, oriented to cancel the camera motion with respect to the 3D GRU memory state $m_{t-1}$ built thus far, and then used to update the 3D GRU memory state. The memory map is projected to specific viewpoints and decoded into a corresponding RGB image.

and occlusion configurations (right). Often, it would not even make sense to label objects in such images, as most objects appear only half-visible. In the case of Internet vision, the picture is the **output** of visual perception of a well-trained visual agent, the human photographer; while for mobile robotic vision, the picture is the **input** to such visual perception. Thus, different architectures may be needed for each.

We present Geometry-aware Recurrent Neural Network architectures, which we call GRNNs, that learn to "lift" 2D image features into 3D feature maps of the scene, while stabilizing against the egomotion of the agent. They are equipped with a 3-dimensional latent feature state: the latent feature vectors are arranged in a 3D grid, where every location of the grid encodes a 3D physical location in the scene. The latent state is updated with each new input frame using egomotion-stabilized features, as shown in Figure 2. GRNNs learn to map 2D input visual features to a 3D latent feature map, and back, in a differentiable manner. To achieve such differentiable and geometrically-consistent mapping between the world scene and the 3D latent feature space, they are equipped with differentiable geometric operations, such as egomotion stabilization, 3D-to-2D projection, and 2D-to-3D unprojection. Beyond being space-aware, we do not impose any other constraints on the learned representations: they are free to encode whatever is relevant for the downstream task.

We train GRNNs in a self-supervised manner to predict image views from novel camera viewpoints, given short frame sequences as inputs. We empirically show GRNNs learn to predict novel views and **strongly generalize** to novel scenes with different number, appearances and configuration of objects. They greatly outperform geometry-unaware networks of previous works that are trained under the exact same view-prediction loss, but do not use egomotion-stabilized convolutions or a 3D latent space. We argue strong generalization is a necessary condition for claiming the ability to spatially reason. Furthermore, the resulting representations support scene arithmetic: adding/subtracting latent scene representations, and decoding the resulting representation from a particular viewpoint, matches the result of adding/subtracting 3D world scenes directly.

## 2 GEOMETRY-AWARE RECURRENT NETWORKS

The proposed GRNNs are recurrent neural networks whose latent state $m \in \mathbb{R}^{w \times h \times d \times c}$ learns a 3D deep feature map of the visual scene. We use the terms 4D tensor and 3D feature map interchangeably, to denote a set of feature channels, each being 3-dimensional. The memory map is updated with each new camera view in a geometrically-consistent manner, so that information from 2D pixel projections that correspond to the same 3D physical point end up nearby in the memory tensor, as illustrated in Figure 2 (right). This permits later convolutional operations to have a correspondent input across frames, as opposed to it varying with the motion of the observer. We believe this is a key for generalization. The main components of GRNNs are illustrated in Figure 2 (right) and are detailed right below.

**Unprojection**    At each timestep, we feed the input RGB image $I$ to a 2D convolutional encoder-decoder network with skip-connections (2D U-net (6)) to obtain a set of 2D feature maps $\mathcal{F} \in \mathbb{R}^{w \times h \times c}$. We then unproject all feature maps to create a 4D feature tensor $V \in \mathbb{R}^{w \times h \times d \times c}$ as follows: For each "cell" in the 3D feature grid indexed by $(i, j, k)$, we compute the 2D pixel location $(x, y)$ which the center of the cell projects onto, from the current camera viewpoint:

$$[x, y] = [f \cdot i/k, f \cdot j/k],$$

where $f$ is the focal length of the camera. Then, $V_{i,j,k,:}$ is filled with the bilinearly interpolated 2D feature vector at that pixel location $(x, y)$. All voxels lying along the same ray casted from the camera center will be filled with nearly the same image feature vectors. The unprojected tensor $V$ enters a 3D encoder-decoder network with skip connections (3D U-net) to produce a resulting feature tensor $\bar{V} \in \mathbb{R}^{w \times h \times d \times c}$.

**Egomotion stabilization and recurrent map update**    Next, we orient the tensor $\bar{V}$ to cancel the relative rotation $\bar{r}$ with respect to our 3D memory map $m_{t-1}$, we denote the oriented tensor as $\bar{V}'$. Once the feature tensor has been properly oriented, we feed $\bar{V}'$ as input to a 3D convolutional Gated Recurrent Unit (3) layer, **whose hidden state is the memory map** $m \in \mathbb{R}^{w \times h \times d \times c}$, as shown in Figure 2 (right). The hidden state is initialized to zero at the beginning of the frame sequence. For our view prediction experiments (Section 3) we found that when the number of views is fixed to ($T = 4$), then average pooling ($m = \frac{1}{N} \sum \bar{V}'$) works equally well to using the GRU update equations, while being much faster.

**Projection and decoding**    Given a 3D feature state $m$ and a desired viewpoint $q$, we first rotate the 3D feature map so that its depth axis is aligned with the query camera axis. We then generate for each depth value $k$ a corresponding projected feature map $p_k \in \mathbb{R}^{w \times h \times c}$. Specifically, for each depth value, the projected feature vector at a pixel location $(x, y)$ is computed by first obtaining the 3D location it is projected from and then inserting bilinearly interpolated value from the corresponding slice of the 4D tensor $m$. In this way, we obtain $d$ different projected maps, each of dimension $w \times h \times c$. Our $d$ depths range from $D - 1$ to $D + 1$, where $D$ is the distance to the center of the feature map, and are equally spaced. Note that we **do not attempt to determine visibility of features at this projection stage**. The stack of projected maps is processed by 2D convolutional operations and is decoded using a residual convLSTM decoder, similar to the one proposed in (4), to an RGB image. We do not supervise visibility directly. The network **implicitly learns to determine visibility** and choose appropriate depth slices from the stack of projected feature maps.

## 3  EXPERIMENTS

We consider the following simulation datasets: i) *ShapeNet arrangement* from  (2) which consists of scenes that have synthetic 3D object models from ShapeNet (1). We follow the same train/test split of ShapeNet (1) so that object instances which appear in the training scenes do not appear in the test scenes. Each scene contains two objects, and each image is rendered from a viewing sphere which has $3 \times 18$ possible views with 3 camera elevations $(20°, 40°, 60°)$ and 18 azimuths $(0°, 20°, \ldots, 340°)$. There are 300 different scenes in the training set and 32 scenes with novel objects in the test set. ii) *Shepard-metzler* shapes dataset from  (4). It contains scenes which consist of seven colored cubes stuck together in random arrangements.  iii) *Rooms-ring-camera* dataset from  (4), a random rooms environment consisting of random floor and wall colors and textures, and variable numbers of shapes in each room of different geometries and colors.

We compare the proposed GRNNs against the recent "tower" architecture of Eslami et al. (4), a 2D network trained under a similar view prediction loss, that has a 2D instead of 3D feature space, and no egomotion-stabilized convolutions. The tower architecture takes as input each 2D image and performs a series of convolutions on it. The camera pose from which the image was taken is tiled on the width and height axes and then concatenated into the feature map after the third convolution. Finally, the feature maps from all views are combined via average pooling. Both our model and the baseline use the same autoregressive decoder network.

Test results are shown in Figure 3. On the left of Figure 3, the distribution of the test scenes matches the training scene distribution. Our model outperforms the baseline in visual fidelity. On the right in Figure 3, the test scene distribution does not match the training one: we test our model and baseline on **scenes with four objects, while both models are trained on scenes with exactly**

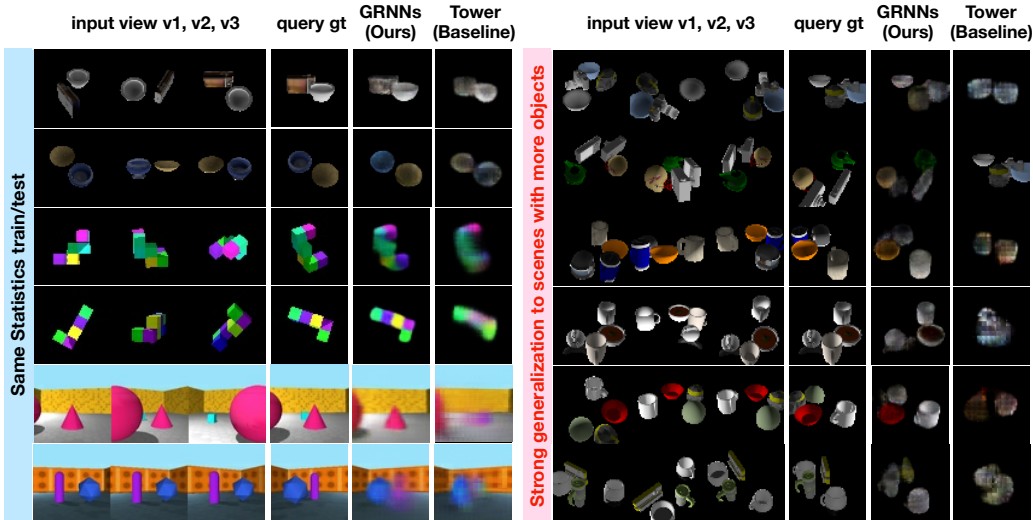

Figure 3: **View prediction results** for the proposed GRNNs and the tower model of Eslami et al. (4). On the left, we show results from the ShapeNet arrangement test set of (2) and the Shepard-Metzler and Rooms-ring-camera datasets of (4). On the right, we show test results on scenes with four objects from the ShapeNet arrangement dataset. While both models were trained only on scenes with two objects, GRNNs outperform the baseline by a large margin and *strongly* generalize under a varying number of objects.

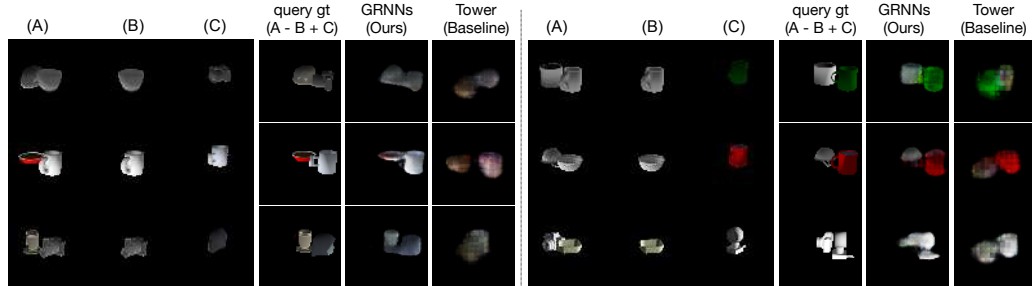

Figure 4: **Scene arithmetic** from the proposed GRNNs and the model of Eslami et al. (4) (tower). Each row is a separate "equation". We start with the representation of the scene in the leftmost column, then subtract (the representation of) the scene in the second column, and add the (representation of the) scene in the third column. We decode the resulting representations into an image view. The groundtruth image is shown in the forth column. It is much more visually similar to the prediction of GRNNs than to the tower baseline.

**two objects.** In this case, our model shows *strong generalization* and outperforms by a margin than our geometry-unaware baseline of (4). Indeed, the ability for spatial reasoning should not be affected by the number of the objects present in the scene. The results above suggest that geometry-unaware models may be merely memorizing views with small interpolation capabilities, as opposed to learning to spatially reason. We attribute this to their inability to represent space efficiently in their latent vectors, a problem the proposed architectures correct for.

**Scene arithmetics** In Figure 4, we show the learnt representations of GRNNs under view prediction are capable of scene arithmetic. The ability to add and subtract individual objects from 3D scenes just by adding and subtracting their corresponding latent representations demonstrates that our model disentangles what from where. In other words, our model learns to store object-specific information in the regions of the memory which correspond to the spatial location of the corresponding object in the scene. Therefore, it is relatively straightforward to carry out scene arithmetic with our model.

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
