# OpenReview forum: "Learning Spatial Common Sense with Geometry-Aware Recurrent Networks"
_ICLR.cc/2019/Workshop/LLD — LLD 2019_

### Official Review · AnonReviewer1 · 2019-04-07
**Good design, great results**

**Rating:** 4
**Confidence:** 2

**Review:**

Summary:
The authors design an architecture for building a geometry-aware latent representation of a scene, and test this by querying a view from the latent representation and comparing it with the actual view.

The paper is structured well, it is clear on what the authors would like to say. Recently geometry-aware architectures have gained importance, and the task for which the authors have shown results is promising.

The authors designed a geometry-aware deep neural network, and trained it on multiple (but limited) viewpoints of multiple scenes to make a 3D latent representation of each scene. They then query a novel viewpoint of each scene and make a prediction loss on that to train the network in a self-supervised manner.

Although it is "self-supervised", it is not clear whether the "novel" viewpoints used while training are within the training set, because if they are not, then the training data is actually as big as the number of iterations used to train. It would be helpful if the authors could clarify on this.

The authors show that their architecture is able to give quality results for scenes with more objects than those on which the network was trained, which provides evidence to the generalizability of the network. This is a very good result, provided we are clear on the "limited"-ness of the training data.

The authors have compared their results with those of another recent architecture that tried to tackle the same problem. The results seem to be in favour of this paper, especially in the case of more objects in the scene. It is worth noting that the other method was not geometry-aware by design, as this paper is.

---

### Official Review · AnonReviewer2 · 2019-04-07
**Article evaluates an embedding of 2d-images that has learnt to predict the change in viewpoint (Rotation+translation) of 3D surfaces. The learning is done on synthetic 3D datasets.**

**Rating:** 4
**Confidence:** 2

**Review:**

Interesting article on evaluating camera's egoview based embedding of images into a 3D latent space that can reconstruct the 3D viewpoint of surfaces in the scene. Article quite short and dense and difficult to understand all the details.

- How does the 3D-Unet evaluate the Ego-motion of the camera viewpoint ? Given the article is short, explanation on the architecture are quite fundamental.
- Does the 3D latent space resemble any shape or surface ? Or is the latent space representation abstract ?
- How are the intrinsic and extrinsic parameters of the camera evaluated in this setup ? We require these parameters to be estimated when evaluating a homography between two views.
- Though the number of labels during training required are small, we still require a rich set of 3D object datsets and different viewpoints generated from them. Can you provide an idea of how this architecture generalizes to new classes and shapes of objects in images.
- An idea of memory consumption for such architectures would be quite useful.

---

### Decision · Program_Chairs · 2019-04-08
**Acceptance Decision**

Accept